# Combination of Cinnamaldehyde with Carvacrol or Thymol Improves the Mechanical Properties of Tibia in Post-Peak Laying Hens

**DOI:** 10.3390/ani12223108

**Published:** 2022-11-10

**Authors:** Huaiyong Zhang, Yongshuai Wang, Yilu Wang, Leilei Wang, Xiangyun Lv, Guangya Cui, Longxiang Ji, Yanqun Huang, Joris Michiels, Wen Chen

**Affiliations:** 1College of Animal Science and Technology, Key Laboratory of Animal Biochemistry and Nutrition, Ministry of Agriculture, Henan Agricultural University, Zhengzhou 450002, China; 2Laboratory for Animal Nutrition and Animal Product Quality, Department of Animal Sciences and Aquatic Ecology, Ghent University, 9000 Ghent, Belgium; 3Charoen Pokphand Group, Zhumadian 463000, China

**Keywords:** essential oils, tibia, intestinal barrier, inflammation, laying hens

## Abstract

**Simple Summary:**

In the current study, the roles of dietary cinnamaldehyde, carvacrol, and thymol blend in the tibia characteristics in post-peak laying hens were evaluated. We firstly analyzed the tibia bone properties and demonstrated that a diet supplemented with a 100 mg/kg combination of cinnamaldehyde with carvacrol or thymol could increase the strength of tibia in layers. We also found that the role of dietary essential oils enhanced intestinal barrier, decreased systemic inflammation, and reduced bone resorption marker in serum. These data indicated that a diet with a blend of cinnamaldehyde and carvacrol or thymol could improve the mechanical properties of tibiae for laying hens, in which enhancing intestinal barrier and decreasing systemic inflammation might be a key mediator.

**Abstract:**

Roles of plant-derived cinnamaldehyde, carvacrol, and thymol in the gut and bone health of laying hens was evaluated in the present study. After acclimation for 2 weeks, a total 384 of 52-week-old laying hens were allocated into three groups for 6 weeks: (1) basal diet group (Ctrl), (2) combination of cinnamaldehyde with carvacrol group (CAR+CIN), and (3) blend of cinnamaldehyde with thymol (THY+CIN). The dietary essential oil level was 100 mg/kg. Each treatment group had eight replicate pens (16 birds/pen). The stiffness and ultimate load of the tibiae from both the CAR+CIN and THY+CIN groups were higher than that of the Ctrl group (*p* < 0.05), along with comparable tibia ash, calcium, and phosphorus content among groups. At the same time, the manipulation of essential oils upregulated the transcription abundances of intestinal barrier proteins to varying degrees, whereas the experimental treatment failed to affect the composition in phyla of cecal microbiota. When compared to the Ctrl group, birds fed the CAR+CIN and THY+CIN diet displayed decreased bone resorption markers, reduced interleukin-1 concentrations, and increased transforming growth factor beta levels in serum. These findings suggest that cinnamaldehyde with carvacrol or thymol in feed of hens could enhance intestinal barrier and improve the mechanical properties of tibiae through structural modelling but not increase the mineral density, which might be involved in suppressing inflammation-mediated bone resorption.

## 1. Introduction

The awareness concerning animal welfare in domestic birds has led to increased attention to osteoporosis, an endocrine disease characterized by bone loss and deterioration of bone microstructure, resulting in increased bone fragility and fracture [1,2]. The prevalence can be especially high in caged laying hens. The restricted movement, lack of exercise, and the calcium (Ca) demand for eggshell production make laying hens prone to osteoporosis. It was reported that the average percentage of laying hens with keel bone fractures was 40.0% at 37 weeks of age [3], 54.4% at 42 weeks of age [4], and 62.0% at 60 weeks of age [5] in furnished cages.

Although the bone growth, mineralization, and mechanical properties depend on the genetic program, these features are also modified by dietary mineral concentration, hormone, cytokines, etc., some of the most important of which are the pro-inflammatory cytokines, such as tumor necrosis factor-alpha (TNF-α), interleukin-6 (IL-6), and IL-1; these cytokines largely contribute to the osteoclastogenesis through interaction with nuclear factor kappa B receptor activating factor ligand (RANKL) in osteoclast precursor cells [6]. The binding of RANKL to its receptor activates multiple downstream signaling pathways, including those of nuclear factor kappa B (NF-κB), to induce the activation of transcription factors for osteoclastogenesis and results in the expression of osteoclast-specific genes including cathepsin K, integrin β3, and tartrate-resistant acid phosphatase (TRAP) [6,7]. Data from mice with TNF-α induced arthritis showed that TNF-α could stimulate the proliferation and/or differentiation of osteoclast precursors [8]. In addition, IL-6 deficiency could reverse bone loss when compared to the estrogen-injected mice [9]. From this perspective, inhibition of pro-inflammatory reactions could be an effective treatment for osteoporosis by interfering with osteoclast differentiation and bone resorption mediated by activated osteoclasts.

In addition to the transport of water, ions, and macromolecules across the intestinal epithelium, tight junction proteins (TJPs) also play critical roles in protecting the host against paracellular bacterial infiltration and penetration of toxic substrates. Impaired gut integrity was associated with increased systemic inflammation, thereby eliciting osteoclastic bone resorption [10]. Enhancing intestinal epithelial integrity was shown to save trabecular bone caused by *Salmonellas* infection in chickens [11]. Analogously, administration of 60 mg/kg zinc glycine was shown to improve intestinal integrity and suppress bone resorption in meat ducks [12]. Compromise of the intestinal barrier of laying hens fed a diet containing fumonisins was linked with inferior bone mechanics [13]. In addition, alterations in gut microbiota were related to bone remodeling by modifying immunoreaction, hormone secretion, Ca absorption, etc. [10,14]. For example, compared to conventionally raised mice, germ-free mice displayed increased bone mass [15] and decreased level of osteoclasts number, IL-6, RANKL, and TNF-α in osseous tissue [16,17]. Outcomes from our recent study demonstrate that the changes of intestinal integrity and cecal microbiota induced by heat stress in broilers led to reduced bone quality. Subsequently, dietary 25-hydroxycholecalciferol reversed the heat stress-induced bone loss through enhancing the intestinal barrier and suppressing both inflammatory reactions and bone resorption [18]. Moreover, in another study, we also noticed that improved intestinal integrity and gut microbiota by acidification of drinking water could decrease systemic inflammation and bone resorption and consequently improve tibial properties in broilers [19]. These data highlight the importance of the interaction of intestinal barrier and inflammation in bone metabolism.

Several nutritional strategies have been used to improve bone health of domestic birds. Among them, natural compounds have received considerable attention because of their antibacterial and anti-inflammatory effects that can modulate the host inflammatory response [20]. Of particular importance are cinnamaldehyde, carvacrol, and thymol. Due to their antimicrobial and antioxidant properties, cinnamaldehyde, carvacrol, and thymol were shown to improve villi structure and microbiota composition (e.g., increasing *Lactobacilli* and reducing enterococci and *Escherichia coli*) to strengthen the gut health of livestock [2,21,22]. It is suggested that the blend of cinnamaldehyde, carvacrol, and thymol probably enhance bone properties of laying hens through the “gut-bone” axis. To support this, recent studies demonstrate that carvacrol could mitigate osteoclastogenesis by impairing the NF-κB pathway and apoptosis in mature osteoclasts [23]. Gavage with daily 75 mg/body weight (BW) cinnamaldehyde significantly reduced the severity of arthritis, bone erosion and destruction, as well as the level of serum IL-6 induced by collagen in rats [24]. Thymol has also been proven to inhibit RANKL-induced osteoclastogenesis in RAW264.7 and LPS-induced bone loss in mice [25].

In this context, the objective of this study was to evaluate whether a phytogenic feed additive based on the essential oils carvacrol, thymol, and cinnamaldehyde could be a potential nutritional strategy to improve tibia quality, as well as intestinal integrity, gut microbiota, and inflammation status of laying hens.

## 2. Materials and Methods

### 2.1. Ethics Statement

All animal specimen sampling procedures were approved by the Henan Agricultural University Experimental Animal Committee (approval number: HN20210012).

### 2.2. Preparation of Essential Oils

The essential oils were provided by Charoen Pokphand Group (Henan, China). Carvacrol (13.5), cinnamaldehyde (4.5%), holly oil (2.0%), and silicon dioxide (80%) were prepared for the formulation of CAR+CIN. Thymol (13.5%), cinnamaldehyde (4.5%), and silicon dioxide (82%) were formulated for the THY+CIN essential oil.

### 2.3. Animal and Study Design

Fifty-week-old Hy-Line Brown laying hens with similar BW (1.77 ± 0.14 kg) were randomly distributed into 3 groups with 8 cages of 16 birds each, i.e., the Ctrl group (fed with a basal diet), the CAR+CIN group, and the THY+CIN group. Two weeks of acclimatization was allowed to the hens, and they were fed the same basal diet to satisfy the nutrient requirements [26], including 3.6% Ca and 0.3% available phosphorus (P) (Table 1). The essential oils were thoroughly mixed with the basal diet. Diets were supplemented in a mashed form to avoid the deactivation of essential oils. The experimental period lasted from 52 weeks of age until 58 weeks of age when the experimental diets were given. Birds were kept in 4-layer vertical cages and housed in a temperature-controlled room with a lighting schedule of 16 h of light and 8 h of darkness. Average ambient temperature and relative humidity were kept at 23 to 25 °C and 30% to 50%, respectively. Throughout the entire trial, the mash diets and water were provided ad libitum.

### 2.4. Sample and Data Collection

The birds were weighed at the end of 52 and 58 weeks. Feed intake was measured weekly for each cage. The egg production and egg weight were recorded daily on a replicate basis, and average egg mass and the ratio of total feed intake and total egg weight were calculated during the period from 52 to 58 weeks. After the 6-week period, one bird from each replicate was randomly selected for weighing and sampling. Blood was collected from the jugular vein after fasting for 8 h. Serum was prepared after centrifugation and stored at −80 °C until analysis. Thereafter, the left tibia was dissected immediately without soft tissues and weighed for the calculation of relative tibia fresh weight after being dried by filter paper. Tibial length and mean diameter of the tibia at mid-diaphysis (because the mid-diaphysis of bird is generally of an elliptical shape) were determined using a Vernier caliper. The circumference of the middle part of tibia were measured by flexible rule. Whereafter, these tibiae were used for analysis of bone mineral content and biomechanical properties. The right tibia and mid-ileum (around 1 cm) were collected for histology analysis. The mid-ileal mucosa and cecal contents were examined for gene expression and microbiome determination, respectively. 

### 2.5. Mechanical Testing 

The mechanical property of tibiae was tested by the three-point bending method using the texture analyzer (TA.XT. Plus, Stable Microsystems Ltd., Godalming, UK). Bone was supported on two supports separated by a 30 mm distance. A 490 N load cell with 5 mm/min rate was employed to load on the bone anterior aspect until breaking of the bone. Force-displacement data were collected, and stiffness, yield load, ultimate load, and the area under the curve (AUC) were calculated, as in a previously described method [27]. 

### 2.6. Tibia Ash Concentration

The tibia was extracted by refluxing diethyl ether in a Soxhlet apparatus for 16 h, oven-dried at 108 °C for 24 h for fat-free bone weight (g) determination. Subsequently, the dry-defatted tibiae were ashed in a muffle furnace at 550 °C for 24 h, and the ash content was calculated and expressed as the percentage of dry-defatted weight.

### 2.7. Serum Biochemical Analysis

Serum immunoglobulins (Ig), including IgG, IgA, and IgM, were quantified separately with the commercial chicken-specific enzyme-linked immunosorbent assay (ELISA) kits, in which the serum samples were diluted 1:125,000 for IgG determination or 1:10,000 for both IgA and IgM determination using phosphate-buffered saline (PBS). The levels of TNF-α, IL-1, IL-6, IL-10, and transforming growth factor beta (TGF-β) in serum were measured with commercial kits. In addition, serum procollagen type I N-terminal propeptide (P1NP) and C-terminal cross-linked telopeptide of type I collagen (CTx) were measured using commercial assay kits in accordance with the manufacturer’s instructions. All kits were obtained from Meimian Industrial Co., Ltd. (Jiangsu, China). All samples were tested in triplicate within each assay.

### 2.8. Intestine and Tibia Histomorphological Analysis

Formalin-fixed ileal samples were dehydrated, embedded, sliced into 5-μm transects, and stained with hematoxylin and eosin (H&E), and subsequently villus height and crypt depth of at least 10 well-oriented villi were measured, and the ratio of the villus height to the crypt depth was calculated. In addition, to visualize bone resorption, the fixed proximal tibia samples were decalcified in 14% EDTA (pH 7.4) for 21 d, embedded in paraffin, longitudinally sectioned into 10-μm slices, and subjected to TRAP bone staining using the leukocyte acid phosphatase assay kit (Sigma-Aldrich, Shanghai, China) according to the instructions.

### 2.9. Cecal Microbiota Composition

Six samples of cecal content from each dietary treatment were collected for DNA extraction. The V4 region of the 16S rRNA gene were amplified using specific primers 515F (5′-GTGCCAGCMGCCGCGGTAA-3′) and 806R (5′-GGACTACHVGGGTWTCTAAT-3′) with 12 nucleotide unique barcode. After mixture and purification, PCR products were sequenced using the Illumina HiSeq platform (Novogene Bioinformatics Technology Co., Ltd., Beijing, China). Low-quality reads were filtered (q < 30), and potential chimeric sequences were removed using the Uchime algorithm. After finding duplicate sequences, all the singletons were discarded due to their possible bad amplicons, which may lead to an overestimation of diversity. Sequences were clustered into operational taxonomic units at 97% identity threshold based on the UPARSE algorithm in USEARCH (v7.0.1090). Taxonomy was assigned using the SILVA database (v1.32) and uclust classifier in QIIME with default parameters. The relative abundance of the taxon at phylum level was produced based on the operational taxonomic unit abundance and taxonomic annotation. The Simpson index was calculated using the OTU table in R, and β-diversity metrics were calculated based on unweighted Unifrac distances and visualized using principal component analysis (PCA). 

### 2.10. Quantitation of mRNA Related Intestinal Barrier

Relative quantification of mRNA levels of Zonula occludens-1 (*ZO-1*), *Claudin-1*, *Occludin*, Cadherin 1 (*CDH1*), and *MUC-2* was performed by RT-PCR. Primers were designed using online Primer 3 and are listed in Table 2. Total RNA was extracted from ileal mucosa, the RNA quality (intact ribosomal RNA 28s/18s) was evaluated by agarose gel electrophoresis, and RNA concentrations were quantified using a spectrophotometer (NanoDrop 2000; Thermo Fisher Scientific Inc., Shanghai, China). First-strand complementary DNA (cDNA) was reverse-transcribed from 200 ng of total RNA using the PrimeScript™ RT Reagent Kit (Takara, Dalian, China). The obtained cDNA was amplified by 40 cycles to determine the mRNA expression of genes of interest. Amplification was conducted with denaturation for 15 min at 95 °C, followed by 40 cycles of denaturation for 30 s at 95 °C, and annealing/elongation for 34 s at 60 °C, and a final melting curve analysis. A total of three housekeeping genes (glyceraldehyde-3-phosphate dehydrogenase, ribosomal protein S9, and *β-actin*) were assessed for stability of expression using two separate cDNA from each treatment (data not shown). The β-actin finally was selected as the reference gene to normalize desired gene expression.

### 2.11. Statistical Analysis

Statistical analysis was carried out using GraphPad Prism Version 8.0 (Graph Pad Software, Inc., La Jolla, CA, USA). Data were checked for normality via the Shapiro–Wilk test before analysis of variance (ANOVA) analysis. One-way ANOVA, including post-hoc analysis (Tukey) or Kruskal–Wallis with Dunn’s test for normally or non-normally distributed data, respectively, were used to evaluate the statistical differences of biological parameters. Of note, due to very little cecal content of some hens, only six birds per treatment were used for cecal microbiota analysis. In addition, to determine the correlation between tibia ultimate load and cecal microbiota, Pearson’s correlation analysis was performed, and correlation coefficients were calculated: *p* < 0.05 and *p* < 0.1was defined as statistically significant and tendency, respectively.

## 3. Results

### 3.1. Dietary Essential Oils Did Not Affect Body Weight, Feed Intake, or Egg Production

The final BW at 58 weeks, as well as gain and feed intake during 52–58 weeks of the laying hens among treatment groups were not different (Figure 1A–C). For egg production, no treatments effects were observed regarding laying rate, average egg mass, and the ratio of feed intake to egg mass (Figure 1D–F).

### 3.2. Diet with Cinnamaldehyde and Carvacrol Decreased Tibial Width and Circumference

The administration with essential oils tended to decrease the fresh weight of tibiae when compared to the Ctrl group (*p* = 0.053), even though the relative weight was not statistically different between the groups (*p* = 0.171) (Table 3). The geometrical characteristics of tibiae were also assessed and showed that the length of tibiae was similar among experimental groups (*p* > 0.05). Significant lower bone width was found in the CAR+CIN group when compared to the Ctrl group. The tibia circumference of birds fed the CAR+CIN diet was also lower than those that received the THY+CIN diet. Regarding bone chemical composition, no notable changes were visible in tibia ash content among the three groups (*p* > 0.05).

### 3.3. Mechanical Properties Are Improved by Essential Oils Supplementation

The results of mechanical testing of the tibiae are presented in Figure 2. The stiffness (slope of the linear portion of the load–displacement curve) and ultimate load of the tibiae from both the CAR+CIN and THY+CIN group were higher than that of the Ctrl group (both *p* < 0.05; Figure 2A,B). The bones of the CAR+CIN group had significantly higher yield loads than that of the Ctrl group (Figure 2C). However, the AUC was found to be slightly lower in the CAR+CIN group than in the Ctrl group (*p* = 0.057; Figure 2D).

### 3.4. Histomorphology and Intestinal Barrier of Ileum Affected by Diet with Cinnamaldehyde and Carvacrol

The changes in the ileum of birds were evaluated by H&E staining (Figure 3A). Dietary treatment did not change villus height, whereas the CAR+CIN- and CAR+THY-treated birds exhibited lower crypt depth when compared to Ctrl layers (Figure 3B-C). Consequently, the birds that received the CAR+CIN diet showed a higher ratio of villus height to crypt depth than did the Ctrl hens (Figure 3D). In addition, the effect of essential oils on tight junction-associated mRNA expression levels was evaluated in this study. When compared to the Ctrl group, birds fed the CAR+THY diet displayed significantly higher mRNA levels of ZO-1 and claudin-1 (Figure 3E). Laying hens that consumed the CAR+CIN diet had higher transcription abundance of MUC-2 as compared to Ctrl (Figure 3F). 

### 3.5. Cecal Microbiota Composition Was Not Affected by Diet

The cecal microbial taxonomy was analyzed and is shown in Figure 4. There was no significant difference among all treatments for the Simpson index (Figure 4A). According to the PCA of beta diversity, the cecal microbiota of the essential oils-treated group did not differ from the Ctrl group (Figure 4B). Analysis of OTUs identified that Bacteroidetes and Firmicutes were the most dominant phyla in the cecum of laying hens (Figure 4C), and a diet with essential oils tended to decrease the ratio of Firmicutes to Bacteroidetes (*p* = 0.091), although it failed to change the proportion of both Firmicutes and Bacteroidetes as compared to the Ctrl group (Figure 4E,F). Further, the ratio of Firmicutes to Bacteroidetes has a negative correlation with tibia ultimate load (r = −0.179, *p* = 0.080; Figure 4G).

### 3.6. Dietary Essential Oils Supplementation Decreased Systemic Inflammatory Status 

There were no significant differences between groups regarding the concentration of IgA, IgG, and IgM in the serum (Figure 5A). Dietary supplementation of CAR+CIN significantly decreased the serum pro-inflammatory factor IL-1 level as compared to Ctrl (Figure 5B). When compared to the Ctrl group, the laying hens that consumed THY+CIN notably increased the anti-inflammatory factor TGF-β content in serum (Figure 5C). Dietary essential oils administration failed to change serum IL-6, TNF-α, and IL-10 concentrations in laying hens (Figure 5B,C).

### 3.7. Diet with Essential Oils Inhibited Bone Resorption

TRAP-positive cells from Ctrl birds were distinctly increased, and supplementation of CAR+CIN reduced the number of TRAP-positive cells in proximal tibiae (Figure 6A). The outcome of serum bone turnover marker analysis revealed that the concentration of bone resorption marker CTx was decreased by dietary CAR+CIN treatments when compared with the Ctrl diet (Figure 6B). Serum P1NP level, representing bone formation, was similar among Ctrl, CAR+CIN, and THY+CIN birds (Figure 6C).

## 4. Discussion

The prevalence of locomotion (gait) problems and bone loss in domestic birds increase markedly due to genetic progress and intensive nutrition. To improve the bone quality and decrease the incidence of leg disease of poultry, essential oils may provide potential alternatives. In this study, a diet contained cinnamaldehyde with carvacrol or thymol improved whole-bone bending strength that could be explained by the associated changes in intestinal barrier and inflammation-induced bone resorption.

Bone health of laying hens are of major concern in practice as there is increasing awareness concerning animal welfare and economic losses. The tibia biomechanical characteristics such as tibia breaking strength and bone ash are usually used as indicators of mineral adequacy and bone development. Although there was no apparent change in the content of ash, the results of whole bone mechanical testing of tibiae demonstrated that the bones of layers fed a diet with 100 mg/kg essential oils were mechanically superior in most variables tested, evidenced by higher whole bone stiffness, yield load, and maximal load. Analogously, 2 g/kg feed *Citrullus lanatus* essential oils inclusion in diets was observed to significantly improved tibia ash, weight, and bone strength in layer hens [28]. Diet with mixed essential oils and betaine could reverse the significant reduction in bone mineralization as evaluated by tibia break strength and total ash from the tibiae in broilers subjected to heat stress [29]. It is well-established that bone strength and the consequent risk of fracture are dependent on the interplay between the material and structural properties of the bone. Bones cannot be both very tough and very stiff, and an inverse relationship exists between these properties [30]. In this regard, AUC is a measure of the amount of energy required to cause fracture, and ductile bones require a larger amount of energy to fail than brittle bones. The decreased AUC in essential oil-treated birds suggested their tibiae were stiffer and stronger than those of the Ctrl group. Of note, the outcomes of bone morphological change indicated that dietary cinnamaldehyde with carvacrol decreased tibia width and circumference in this study, which might result from the biased selection of birds. The post-peak laying hens used in this study are mature; the bones may also be mature and do not growth in length and width. The potential treatment-differences mainly occur in medullary bone, such as bone remolding. Therefore, these changes are probably explained by the biased selection during sampling. In combination with the comparable tibia ash level and improved bone mechanical characteristics, it is possible that the birds could adapt their bone morphology and the proportion of organic compound and mineral content within bone to suit the late stages of egg production.

Mineral content is closely associated with the mechanical properties of bone, and improved bone strength was expected to link the increase in bone weight and ash [31]. However, we did not find that the addition of essential oil apparently increased the bone weight, ash, Ca, and P content. Alternatively, the microstructure of tibia changes might be the main contributor to the positive role exerted by the diet containing the cinnamaldehyde with carvacrol or thymol in bone strength [32], although the relevant parameters were not determined in this study. Deformed bone conformations were found to impair gait abnormalities even though the whole bone possessed adequate minerals deposition, especially in proximal metaphysis of the tibia, in which the metabolic processes are the most intensive in the proximal metaphysis, and the cells here are highly sensitive to numerous dietary deficiencies [33]. The link between bone quality and bone remodeling drives us to explore the effects of essential oils on bone formation and resorption. In this regard, gavage administration with cinnamaldehyde (25–75 mg/kg BW/d) was found to linearly increase bone mass of femurs and increase bone resorption in ovariectomized rats, and the presence of cinnamaldehyde (15 and 30 μg/mL) promoted the differentiation of osteoblasts in vitro [34]. Carvacrol treatment (3–15 μg/mL) could linearly reduce the numbers of mature osteoclasts by inhibiting RANKL expression in RAW264.7 macrophages [23]. Moreover, medium with 20 and 40 μM thymol has also been proven to inhibit RANKL-induced osteoclastogenesis in RAW264.7 cells, and oral administration of 100 mg/kg thymol for 10 d restored the LPS-induced bone loss in mice [25]. The anti-osteoclastogenic potential was further supported by the current results that the concentration of bone resorption marker CTx was decreased by the cinnamaldehyde with both carvacrol and thymol. These results suggest that the combination of cinnamaldehyde, carvacrol, and thymol has inhibitory effects on osteoclasts and could be a potential compound for treating osteoporosis and bone abnormalities of laying hens.

Essential oils inhibit osteoclastogenesis and negatively regulate the osteoclast-mediated bone resorption in this study, and this might be consequence of its anti-inflammation through enhancing the intestinal barrier. Osteoclast formation and activity is closely regulated by inflammatory cytokines [18]. In this study, the blend of cinnamaldehyde and carvacrol or thymol decreased serum pro-inflammatory factor IL-1 level, increased anti-inflammatory factor TGF-β content, and declined bone resorption marker CTx concentration to varying degrees. It is likely that supplementation of 100 mg/kg essential oils in feed of laying hens might suppress inflammatory cytokine-stimulated osteoclastic bone resorption, especially the addition of cinnamaldehyde with carvacrol. Administration with cinnamaldehyde (75 mg/kg BW per day) was also noticed to reduced serum RANKL and IL-6 levels in rats apart from remission collagen that induced the severity of arthritis and bone destruction [24]. Accumulating lines of evidence showed that a blend of phytogenic feed additives comprising 5% carvacrol, 3% cinnamaldehyde, and 2% capsicum oleoresin at 100 mg/kg attenuated the production of pro-inflammatory cytokines in visceral adipose tissues by inhibiting toll like receptor 2 (TLR2)- and TLR4-mediated signaling [35]. A study of carvacrol and osteoclastogenesis found that culture media treated with 3–15 μg/mL carvacrol mitigated osteoclastogenesis by impairing the NF-κB pathway and induction of apoptosis in mature osteoclasts [23]. It is well-known that dysfunction of the intestinal barrier is commonly associated with increased inflammatory response in both ileum and bone marrow that further induces osteoclastic bone resorption [18]. The importance of intestinal barrier to bone quality prompts us to explore the effects of essential oils on TJPs. In the present study, the upregulated the expression of *ZO-1*, *Claudin-1*, and *MUC-2* in essential oil groups is in accordance with a previous study showing that the dietary supplementation of 37 mg/kg essential oils (*Lippia origanoides*) increased the tight junction integrity of 42-d-old broilers, showed by lower serum FITC-d that was used as a biomarker to evaluate intestinal permeability [29]. Contrary to the current research, a diet with a 100 mg/kg carvacrol-thymol blend (1:1) did not change the mRNA abundance of TJPs in the jejunum of weaning piglets [21]. These controversial data may be due to the different dose, source, and/or blend of vegetable extracts. It was pointed that enhancing intestinal integrity could restore the bone loss of trabecular bone induced by *Salmonella* in chickens [11], which supported the opinion that enhancing intestinal integrity through dietary cinnamaldehyde with carvacrol or thymol is a possible mechanism to contribute to the tibial quality of laying hens in the current study. Additionally, alterations in gut microbiota were tightly linked with bone remodeling [10,14]. It was pointed out that dietary carvacrol and thymol treatment increased the abundance of beneficial bacteria and decreased the proportion of potentially harmful bacteria such as *Escherichia coli* and *Clostridium* [36]. Considering the anti-inflammatory and antimicrobial effects exerted by cinnamaldehyde in broilers [37], some alterations in gut microbiota are expected. However, in the current study, the diversity and abundance in phyla levels were comparable among the three groups. Nevertheless, we noticed dietary cinnamaldehyde with thymol slightly induced a decrease in the ratio of *Firmicutes* to *Bacteroidetes*. Previous studies on mice using *Lactobacillus* support the beneficial role for the declining *Firmicutes*: *Bacteroidetes* ratio in preventing femoral and vertebral trabecular bone loss [38]. This was further confirmed by Pearson’s correlation analyses in the current study, i.e., tibia ultimate load negatively correlated with the *Firmicutes*: *Bacteroidetes* ratio. Taken together, these data indicated that improved intestinal integrity by dietary cinnamaldehyde in combine with carvacrol or thymol could decrease systemic inflammation and bone resorption and consequently improve tibia mechanical properties in layering hens, which was in line with our recent findings in broilers [19].

## 5. Conclusions

Taken together, a diet containing a 100 mg/kg blend of cinnamaldehyde and carvacrol or thymol could strengthen the tibial mechanical properties of laying hens, largely due to structural modelling rather than the increase in mineral density. During this procedure, the role of dietary essential oils in enhancing the intestinal barrier and decreasing systemic inflammation might be a key mediator of osteoclast-mediated bone resorption. At the same time, discovering the connection between the intestinal barrier and bone health can speed the application of herbal extract targets for prevention and treatment of osteoporosis and leg problems of laying hens.

## Figures and Tables

**Figure 1 animals-12-03108-f001:**
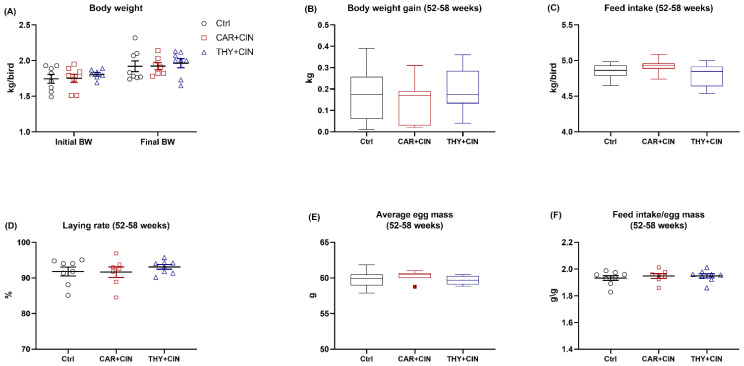
Impacts of the combination of cinnamaldehyde with carvacrol (CAR+CIN) or thymol (THY+CIN) on (**A**) body weight (BW) at week 52 (initial BW) and week 58 (final BW), as well as (**B**) body weight gain, (**C**) feed intake, (**D**) laying rate, (**E**) average egg mass, and (**F**) feed intake/egg mass of laying hens from 52 to 58 weeks. Values are mean ± standard error (SE) represented by vertical bars in a scatter plot. In the box-whiskers plots, boxes are bounded by the 25th and 75th percentiles, with the median shown by the line bisecting the box. Whiskers extend to the full range of the data. Outliers are represented by dots. Statistical significance was identified at *p* < 0.05 (*n* = 8).

**Figure 2 animals-12-03108-f002:**
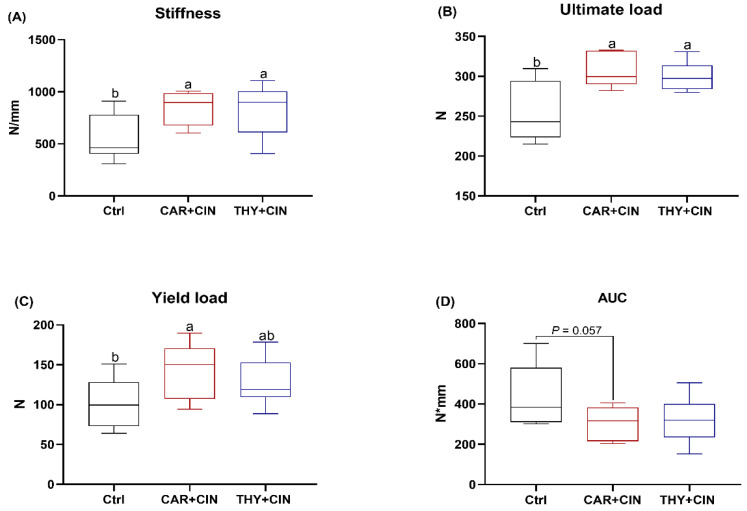
Impacts of the combination of cinnamaldehyde with carvacrol (CAR+CIN) or thymol (THY+CIN) on tibia mechanical properties of laying hens: (**A**) stiffness (N/mm); (**B**) yield point (N); (**C**) ultimate load (N); (**D**) area under the load–displacement curve (AUC, work to failure, N*mm). Boxes are bounded by the 25th and 75th percentiles, with the median shown by the line bisecting the box. Whiskers extend to the full range of the data. ^a,b^ Values without common superscript are notably different (*p* < 0.05, *n* = 8).

**Figure 3 animals-12-03108-f003:**
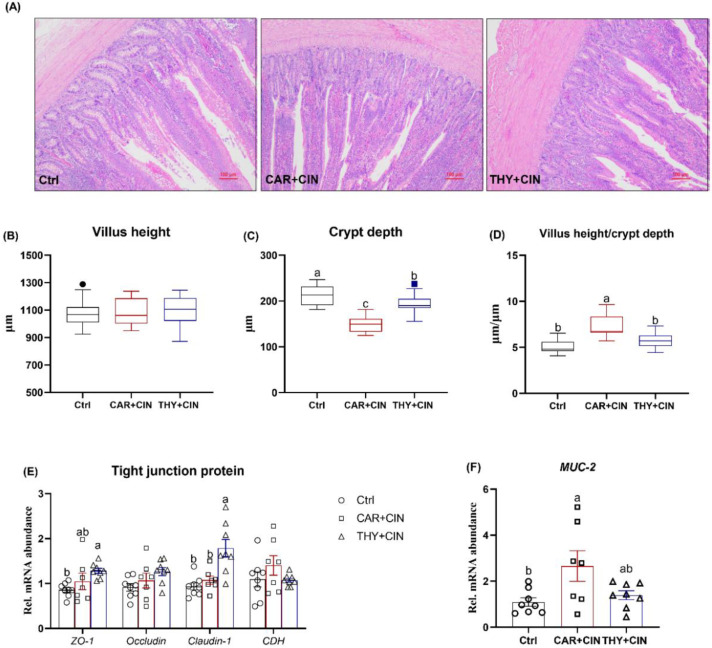
Impacts of the combination of cinnamaldehyde with carvacrol (CAR+CIN) or thymol (THY+CIN) on morphology and intestinal barrier of mid-ileum in laying hens: (**A**) representative hematoxylin/eosin staining (scale bar = 100 μm), and (**B**) villus height, (**C**) crypt depth, and (**D**) the ratio of villus height to crypt depth were measured in ileum, (**E**,**F**) mRNA abundance of Zonula occludens-1 (ZO-1), Occudin, Claudin-1, Cadherin 1 (CDH1), and MUC-2. Boxes are bounded by the 25th and 75th percentiles, with the median shown by the line bisecting the box. Whiskers extend to the full range of the data. Outliers are represented by dots. In the scatter plot, values are mean ± standard error (SE) represented by vertical bars. ^a,b^ Values without common superscript are notably different (*p* < 0.05, *n* = 8).

**Figure 4 animals-12-03108-f004:**
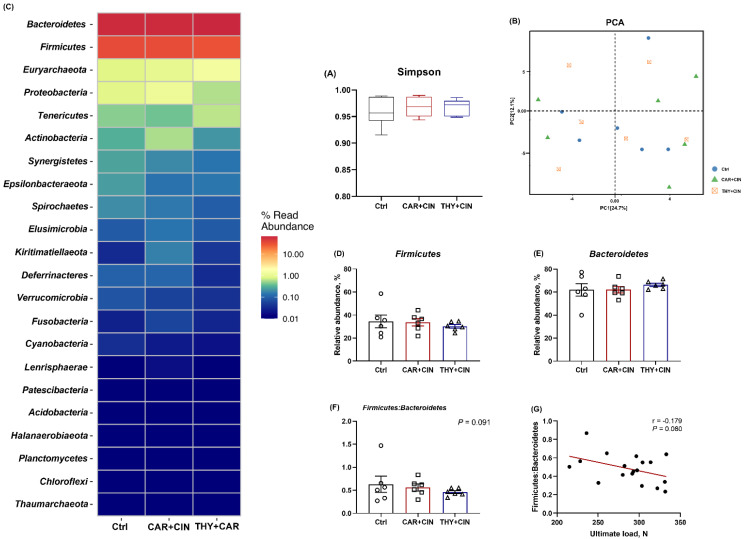
Impacts of the combination of cinnamaldehyde with carvacrol (CAR+CIN) or thymol (THY+CIN) on cecal microbiome of laying hens. (**A**) Simpson indexes was used to assess alpha diversity, in which boxes are bounded by the 25th and 75th percentiles, with the median shown by the line bisecting the box. (**B**) principal component analysis (PCA) of caecum microbiome diversity, (**C**) relative abundances at phylum level, (**D**–**F**) the proportion of Firmicutes and Bacteroidetes, and their ratio, (**G**) correlation between Firmicutes: Bacteroidetes ratio and ultimate load. Values are mean ± standard error (SE) represented by vertical bars (*n* = 6), and *p* < 0.05 is defined as statistically significant.

**Figure 5 animals-12-03108-f005:**
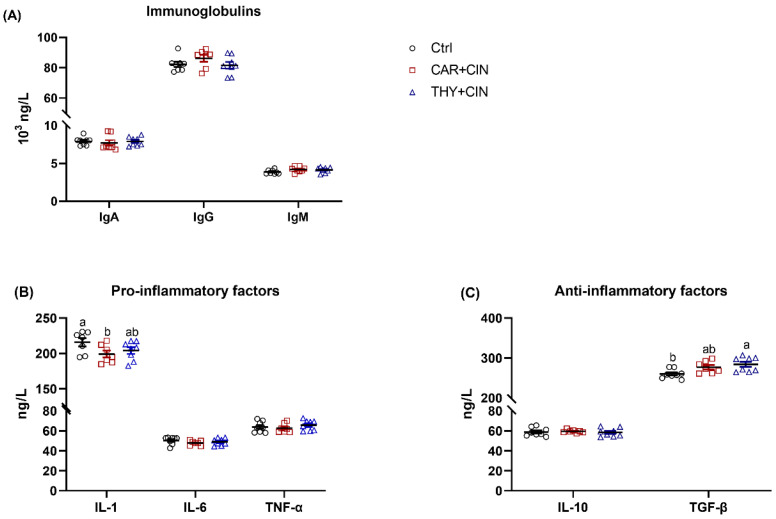
Impacts of the combination of cinnamaldehyde with carvacrol (CAR+CIN) or thymol (THY+CIN) on inflammatory status of laying hens: (**A**) immuneglobulins; (**B**) the pro-inflammatory factors interleukin (IL)-1, IL-6, and tumor necrosis factor alpha (TNF-α); (**C**) the anti-inflammatory factors IL-10 and transforming growth factor beta (TGF-β). Values are mean ± standard error (SE) represented by vertical bars (*n* = 7–8). ^a,b^ Values without common superscript are notably different (*p* < 0.05).

**Figure 6 animals-12-03108-f006:**
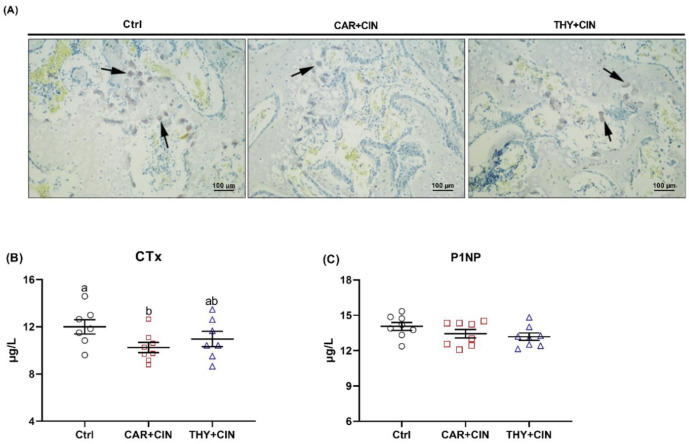
Impacts of the combination of cinnamaldehyde with carvacrol (CAR+CIN) or thymol (THY+CIN) on bone turnover of laying hens: (**A**) tartrate-resistant acid phosphatase (TRAP) staining of tibia sections, (**B**) bone resorption biomarker C-terminal cross-linked telopeptide of type I collagen (CTx) content, and (**C**) bone formation biomarker Procollagen type I intact N-terminal propeptide (P1NP) concentration were determined. Values are mean ± standard error (SE) represented by vertical bars (*n* = 7–8). ^a,b^ Values without common superscript are notably different (*p* < 0.05).

**Table 1 animals-12-03108-t001:** Ingredients and calculated analysis of nutrient in the basal diet (as-fed).

Item	Proportion (%)	Calculated Analysis	Proportion (%)
Corn	32.95	AME, MJ/kg	2620
Wheat	30.0	CP	16.30
Soybean meal (46% CP)	12.01	Calcium	3.60
Wheat bran	3.7	Total phosphorus	0.52
Sprayed corn bran	2.5	Available phosphorus	0.30
Corn gluten	1.5	Lysine	0.88
DDGS ^a^	3.5	Methionine + cystine	0.67
Sodium chloride	0.24	Threonine	0.61
Limestone	8.6		
CaHPO_4_	0.43		
Chicken bone meal	2.0		
Soybean oil	0.57		
Premix ^b^	2.0		
Total	100.0		

^a^ DDGS, distillers dried grains with solubles; AME, apparent metabolic energy; CP, crude protein. ^b^ Provided per kilogram of diet: Cu (CuSO_4_·5H_2_O), 8 mg; Fe (FeSO_4_·7H_2_O), 80 mg; Zn (ZnSO_4_·7H_2_O), 80 mg; Mn (MnSO_4_·H_2_O), 80 mg; Se (NaSeO_3_), 0.3 mg; I (KI), 0.7 mg; vitamin A, 2700 IU; vitamin D, 3400 IU; vitamin E, 10 IU; vitamin K, 0.5 mg; thiamine, 2.0 mg; riboflavin, 5.0 mg; pyridoxine, 3.0 mg; vitamin B_12_, 0.007 mg; calcium pantothenate, 10.0 mg; folate, 0.5 mg; biotin, 0.1 mg; nicotinic acid, 30 mg.

**Table 2 animals-12-03108-t002:** The primers for quantitative real-time PCR.

Gene ID	Gene	Primer Sequences (5′→3′)	Product Length, bp
XM_046925214.1	*ZO-1*	F: GAAGAGAGCACAGAACGCAG R: CACTTGTGGCAAGCTGAAGT	123
NM_001013611.2	*Claudin-1*	F: TCTGGTGTTAACGGGTGTGA R: GTCTTTGGTGGCGTGATCTT	117
NM_205128.1	*Occludin*	F: CGTTCTTCACCCACTCCTCC R: CCAGAAGACGCGCAGTAAGA	107
NM_001039258.3	*CDH1*	F: AGCCAAGGGCCTGGATTATG R: GATAGGGGGCACGAAGACAG	157
NM_001318434.1	*MUC-2*	F: AGTGGCCATGGTTTCTTGTC R: TGCCAGCCTTTTTATGCTCT	80
NM_205518.1	*β-actin*	F: GTCCACCGCAAATGCTTCTAA R: TGCGCATTTATGGGTTTTGTT	78

*ZO-1*, Zonula occludens-1; *CDH1*, Cadherin 1.

**Table 3 animals-12-03108-t003:** Tibia bone properties of laying hens given to dietary essential oils supplementation.

Item	Treatment			*p*-Value
	Ctrl	CAR+CIN	THY+CIN	
Fresh weight, g	11.34 ± 0.51	10.35 ± 0.82	10.84 ± 0.83	0.053
Relative weight, % body weight	0.60 ± 0.06	0.54 ± 0.04	0.56 ± 0.06	0.171
Length, mm	119.14 ± 3.39	117.75 ± 3.68	117.82 ± 3.69	0.693
Width, mm	8.27 ± 0.27 ^a^	7.85 ± 0.34 ^b^	8.23 ± 0.24 ^ab^	0.020
Circumference, cm	2.61 ± 0.11 ^ab^	2.54 ± 0.05 ^b^	2.69 ± 0.06 ^a^	0.010
Ash, % fat-free weight	58.42 ± 2.36	58.40 ± 0.45	57.54 ± 2.06	0.778

The results expressed as mean and standard error. ^a,b^ Values without common superscript are notably different (*p* < 0.05, *n* = 8). CAR, carvacrol; CIN, cinnamaldehyde; THY, thymol.

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
