# Peer review of "Combination of Cinnamaldehyde with Carvacrol or Thymol Improves the Mechanical Properties of Tibia in Post-Peak Laying Hens"

_animals, 2022, doi:10.3390/ani12223108_

Round 1

Reviewer 1 Report

Dear Authors,

The manuscript requires major revision:

1. The experimental methods should be described more, the chemical characterization of the essential oils extract should be presented in the paper or appropriate references should be provided.  For example L113 – “holly oil”? Is this the same as "holly massage oil" ? more information about this ingredient is definitely needed.

2. References from L84-86 should be removed. It is difficult to find any common link for these studies (vitamins, minerals, organic compound) apart from the Authors names. Extensive self-cations is not necessary. Correct to:" Several nutritional strategies have been used to improve bone health of domestic birds". Similar comment to L247-350 – unnecessary self-citations about ducks and broilers.

3. Four weeks-long period is a very short experimental period for hens. Could you provide any references where such short period for older layers was selected for bone-related studies ?

4. Bone Ca and P content: Results in Table 4: It is not possible to obtain Ca:P ratio such low as 1 for layers ! It should be at least 2 or higher !. Also according to the presented data Ca:P ratio is lower than 1 (for all groups, Ca content is lower than P). Finally, references for applied analytical methods (L170-172) are needed.

5. Measurements of bone circumferences: Studies relating to human archeological data [34] does not justify the measurement of bone circumference in this study. Used hens were mature, after the peak lying period, when bones are also mature and do not growth in length and width. The potential treatment-differences should be located rather inside the bone-diaphysis, in medullary bone, where the resorption. process occurs. That’s why you cannot explain the observed differences. In reviewer’s opinion, the only logical explanation is the biased selection of birds to experimental groups.

Please also find the following observations:

Simple summary is missing. Bacteria names and gene names should be written italic throughout the whole manuscript.

L43 [3] - It is over 20-years old reference. Nutritional requirements or recommendation have been revised to overcome bone fracture prevenance since then. Do you have any information about more recent studies concerning  bone fractures in layers ?

L61-63 the primary role of TJP is to control the transport of water, ions and macromolecules across the intestinal epithelium. Please revise. Also [8] is not a correct reference, TJPs are not discussed in this paper.

L82 Complete  bibliographic data of [18] are available since June, correct.

L109 how experimental additives were applied to basal diet ? Also what type of vehicle was used ?

L118-120 Reference needed.

L145 in what plane the perimeter was measures ? hens tibia mid-diaphysis is generally of elliptical shape.

L155 Please use SI units (N)

L175 what dilutions were used during the analyzes? ELISA Igs’ quantifications generally requires highly diluted samples.

L189 please use anatomically defined plane

L212 “mRNA levels of PROTEINS of intestinal barrier”. Also rephase the whole sentence, as it suggests that mucin-2 is a TJ protein. Please also use a proper gene names (written italic) for proteins – for example:  MUC2 gene for mucin-2 protein.

L236 Not clear. One bird per replicate (n=8) was used for tibia characteristic, while only n=6 birds were selected for cecal microbiota analysis. This should be clearly described there.

L254,L261 the tendency (trend) was not defined in M&M section

Figure 2A – correct “slope” to “stiffness”

Figure 2E,F - What does whiskers represent ? SE or SD ?

L335 could you describe analyzed bone sanction in more detailed manner ? Where it was  located? Epiphysis? Diaphysis ?

L397-398 Remove. TD is an abnormality of the growth cartilage (appearance of non-vascularized and non-mineralized cartilage masses in tibial growth plates), which occurs in fast growing young poultry, not aged laying hens.

L416-148, L433-434. Remove, unnecessary repetition of the introduction.

L441 “increase the tight junction INTEGRITY of 42-d-old broilers”

L464 “current study SHOWING that tibia”

L470-473. Bone mineral content was not presented nor discussed in this manuscript, remove.

Author Response

Responses to the reviewers’ comments

Dear Editors and Reviewers:

We are truly grateful to yours and other reviewers’ critical comments and thoughtful suggestions. Based on these comments and suggestions we have made careful modifications on the original manuscript. All changes made to the text are in red color. We hope the new manuscript will meet your journal’s standard. Below you will find our point-by-point responses to the reviewers’ comments/ questions:
1. The experimental methods should be described more; the chemical characterization of the essential oils extract should be presented in the paper or appropriate references should be provided.  For example, L113 – “holly oil”? Is this the same as "holly massage oil"? more information about this ingredient is needed.

Thanks for your suggestions, we added some experimental methods. More important, the essential used in the study were prepared through pure carvacrol, cinnamaldehyde, holly oil, and thymus. Therefore, we specified the source of essential oils as following: “Two formulations of essential oils were prepared in powder and provided by Charoen Pokphand Group (Henan, China) with feed inclusion of 100 g/ton of feed based on the manufacturer’s recommendations. The carvacrol and cinnamaldehyde essential oil (CAR+CIN) composed of 13.5% carvacrol, 4.5% cinnamaldehyde, 2.0% holly oil, and 80% silicon dioxide; the thymus and cinnamaldehyde essential oil (THY+CIN) composed of 13.5% thymol, 4.5% cinnamaldehyde, and 82% silicon dioxide”

2. References from L84-86 should be removed. It is difficult to find any common link for these studies (vitamins, minerals, organic compound) apart from the Authors names. Extensive self-cations is not necessary. Correct to:" Several nutritional strategies have been used to improve bone health of domestic birds". Similar comment to L247-350 – unnecessary self-citations about ducks and broilers.

We removed these references and revised the sentence. Thanks.

3. Four weeks-long period is a very short experimental period for hens. Could you provide any references where such short period for older layers was selected for bone-related studies?

Indeed, the period of study is a very short for hens, which might be main reason why most of results are not significant, indicating the long period is needed from this point. Initially, we assumed that the rapid decline in egg production and quality at the end of the laying cycle might make the bone tends to be more sensitive to some nutrients especially Ca, thus we designed this experiment. Therefore, the significant results from this study showed they are highly susceptible, and the longer period is needed for bone-related studies. We will extent the test period in our future research. We deeply appreciate your professional suggestions.

4. Bone Ca and P content: Results in Table 4: It is not possible to obtain Ca:P ratio such low as 1 for layers! It should be at least 2 or higher! Also, according to the presented data Ca:P ratio is lower than 1 (for all groups, Ca content is lower than P). Finally, references for applied analytical methods (L170-172) are needed.

As for the referee’s concern, we carefully checked our raw data, but we not figured out some wrong happened about the content of Ca. In fact, the ration of bone Ca and P should be at least 2 or higher, indicating that our data are unreasonable. Considering the precision, we decided to delete the related data and methods.

5. Measurements of bone circumferences: Studies relating to human archeological data [34] does not justify the measurement of bone circumference in this study. Used hens were mature, after the peak lying period, when bones are also mature and do not growth in length and width. The potential treatment-differences should be located rather inside the bone-diaphysis, in medullary bone, where the resorption. process occurs. That’s why you cannot explain the observed differences. In reviewer’s opinion, the only logical explanation is the biased selection of birds to experimental groups.

The cite might be improper in this point. More important, reviewer’s opinions are totally correct, based on your professional suggestions, we have made careful modifications on the original manuscript as following: “Of note, the outcomes of bone morphological change indicated that dietary cinnamaldehyde with carvacrol decreased tibia width and circumference in this study, which might result from the biased selection of birds. The post-peak laying hens used in this study are mature, the bones may be also mature and do not growth in length and width. The potential treatment-differences mainly occur in medullary bone such as bone remolding. Therefore, these changes probably are explained by the biased selection during sampling”.

Please also find the following observations:

Simple summary is missing.

We added the summary as following: “In this study, we clarified the effects of dietary cinnamaldehyde in combination with carvacrol or thymol on the tibia characters in post-peak laying hens. We firstly analyzed the tibia bone properties and demonstrated that the supplementation of 100 mg/kg cinnamaldehyde with carvacrol or thymol in feed of layers could strength the mechanical properties of tibia. We also found that the role of dietary essential oils addition in enhancing intestinal barrier, de-creasing systemic inflammation, and reducing bone resorption marker in serum. This study strongly suggested that the supplementation of 100 mg/kg cinnamaldehyde with carvacrol or thymol in feed of layers could strength the mechanical properties of tibia, in which enhancing intestinal barrier and decreasing systemic inflammation might be a key mediator”.

Bacteria names and gene names should be written italic throughout the whole manuscript.

We have done corresponding revision in the manuscript according to this comment.

L43 [3] - It is over 20-years old reference. Nutritional requirements or recommendation have been revised to overcome bone fracture prevenance since then. Do you have any information about more recent studies concerning bone fractures in layers?

As for the referee’s concern, it was reported that the average percentage of laying hens with keel bone fractures was 40.0% at 37 weeks of age (Wei et al., 2021), 54.4% at 42 weeks of age (Wei et al., 2020), and 62.0% at 60 weeks of age (Rodenburg et al., 2008) in furnished cages.

Wei HD, Bi YJ, Xin HW et al. 2020. Keel fracture changed the behavior and reduced the welfare, production performance, and egg quality in laying hens housed individually in furnished cages. Poult. Sci. 99:3334–3342.

Wei HD, Pan L, Li C, et al. 2021. Dietary soybean oil supplementation affects keel bone characters and daily feed intake but not egg production and quality in laying hens housed in furnished cages. Front. Vet. Sci. 8:657585.

Rodenburg TB, Tuyttens FAM, De Reu K, et al. 2008. Welfare assessment of laying hens in furnished cages and non-cage systems: an on-farm comparison. Anim. Welf. 17:363–373

L61-63 the primary role of TJP is to control the transport of water, ions and macromolecules across the intestinal epithelium. Please revise. Also [8] is not a correct reference, TJPs are not discussed in this paper.

According to your suggestions, we revised the description and enhanced the discussion about TJP as following: “In addition to the transport of water, ions and macromolecules across the intestinal epithelium, the tight junction proteins (TJPs) also play critical roles in protection the host against paracellular bacterial infiltration and penetration of toxic substrate” …. “It is well-known that dysfunction of the intestinal barrier is commonly associated with an increased the inflammatory response in both ileum and bone marrow that further induced osteoclastic bone resorption [18]. The importance of intestinal barrier in bone quality prompts us to explore the effects of essential oils on TJPs. In this study, the cinnamaldehyde with carvacrol or thymol essential oil diet upregulated the expression of ZO-1, Claudin-1, and MUC-2, which is in accordance with previous study saying that the dietary supplementation of 37 mg/kg essential oils (Lippia origanoides) increased the tight junction integrity of 42-d-old broilers, showed by lower serum FITC-d that was used as a biomarker to evaluate intestinal permeability [29]. Contrary to the current research, diet with a 100 mg/kg carvacrol-thymol blend (1:1) diet did not change the mRNA abundances of TJPs in jejunum of weaning piglets [21]. These controversial data may be due to the different dose, source, and/or blend form of vegetable extracts”.

L82 Complete bibliographic data of [18] are available since June, correct.

We revised the reference as following: Zhang, H., Y, Guo., Z, Wang., Y. Wang Y, Chen B, Du P, Zhang X, Huang Y, Li P, Michiels J, Chen W. Acidification of drinking water improved tibia mass of broilers through the alterations of intestinal barrier and microbiota. Anim Biosci. 2022 Jun;35(6):902-915.

L109 how experimental additives were applied to basal diet? Also, what type of vehicle was used?

The essential oils were thoroughly mixed with the basal diet. Diets were supplemented in a mashed form to avoid the deactivation of essential oils.

The vehicle used in this study is silicon dioxide.

L118-120 Reference needed.

We added the reference.

Standards China Agricultural Industry Standards. Nutrient requirement of meat-type ducks (NY/T 33-2004). Beijing: China Agricultural Industry Standards (2004).

L145 in what plane the perimeter was measures? hens tibia mid-diaphysis is generally of elliptical shape.

Because the tibia mid-diaphysis of bird is generally of elliptical shape, thus the mean diameter of the tibia was determined by a digital caliper.

L155 Please use SI units (N)

We revised the unit of load cell. Thanks.

L175 what dilutions were used during the analyzes? ELISA Igs’ quantifications generally require highly diluted samples.

Serum immunoglobulins (Ig) including IgG, IgA, and IgM were quantified separately with the commercial chicken-specific enzyme-linked immunosorbent assay (ELISA) kits, in which the serum samples were diluted 1:125,000 for IgG determination or 1:10,000 for both IgA and IgM determination using phosphate-buffered saline (PBS)

L189 please use anatomically defined plane

We changed “frontally” into “longitudinally”

L212 “mRNA levels of PROTEINS of intestinal barrier”. Also, rephase the whole sentence, as it suggests that mucin-2 is a TJ protein. Please also use a proper gene name (written italic) for proteins – for example:  MUC2 gene for mucin-2 protein.

It is intestinal barrier proteins.

In addition, we have done corresponding revision according to this comment in whole manuscript.

L236 Not clear. One bird per replicate (n=8) was used for tibia characteristic, while only n=6 birds were selected for cecal microbiota analysis. This should be clearly described there.

Thanks for your suggestions, we specified the reason why only n=6 birds were selected for cecal microbiota analysis as following: “Due to very little cecal content of some hens, thus only 6 birds per treatment were used to cecal microbiota analysis”.

L254, L261 the tendency (trend) was not defined in M&M section.

We defined the tendency in Statistical analyses as following: P < 0.05 was defined as statistically significant. A trend was regarded as P < 0.1

Figure 2A – correct “slope” to “stiffness”

We correct “slope” to “stiffness”. Thanks.

Figure 2E, F - What does whiskers represent? SE or SD?

As for the referee’s concern, Values are mean ± standard error (SE) represented by vertical bars in scatter plot, i.e., Figure 3 E, F.

L335, could you describe analyzed bone sanction in more detailed manner? Where was it located? Epiphysis? Diaphysis?

As you known, the serum bone turnover markers just revealed that these changes in whole bone remolding, including tibia. It is hard to locate the epiphysis or diaphysis. In this regard, more precise parameters are needed to describe analyzed bone sanction in more detailed manner.

L397-398 Remove. TD is an abnormality of the growth cartilage (appearance of non-vascularized and non-mineralized cartilage masses in tibial growth plates), which occurs in fast growing young poultry, not aged laying hens.

Thank you for professional suggestion, we remove related section.

L416-148, L433-434. Remove, unnecessary repetition of the introduction.

We removed related section in revision manuscript. Thanks.

L441 “increase the tight junction INTEGRITY of 42-d-old broilers”

We completed the description, thanks.

L464 “current study SHOWING that tibia”

We revised the sentence as following: “This was further confirmed by Pearson’s correlation analyses in the current study, i.e., tibia ultimate load negatively correlated with the Firmicutes: Bacteroidetes ratio.”

L470-473. Bone mineral content was not presented nor discussed in this manuscript, remove.

Thanks. We remove the related section.

Reviewer 2 Report

The authors investigated the effect of dietary cinnamaldehyde with carvacrol or thymol essential oils on the gut and bone health in post-peak laying hens. They found that cinnamaldehyde with carvacrol or thymol in the feed of hens could improve the mechanical properties of the tibia through structural modelling but not the increase in mineral density, which might involve in suppressing inflammation-mediated bone resorption due to the positive role of dietary essential oils addition in enhancing intestinal barrier. The manuscript is well written and I suggest publication of this nice manuscript.

I have only some comments:

1-Line 222:  Please revise the cycle program

2- Line 196: please add the directions for the primers

3- Please check the writing of bacterial species, should be italic (please see attached file)

4- Please revise the references for unity according to the journal requirements (please see attached file).

5- For other "Minor comments", please see the attached file.

Author Response

The authors investigated the effect of dietary cinnamaldehyde with carvacrol or thymol essential oils on the gut and bone health in post-peak laying hens. They found that cinnamaldehyde with carvacrol or thymol in the feed of hens could improve the mechanical properties of the tibia through structural modelling but not the increase in mineral density, which might involve in suppressing inflammation-mediated bone resorption due to the positive role of dietary essential oils addition in enhancing intestinal barrier. The manuscript is well written, and I suggest publication of this nice manuscript.

We appreciate the reviewer’s comments

I have only some comments:

1-Line 222:  Please revise the cycle program

We supplemented the cycle program as following: “Amplification was conducted with denaturation for 15 min at 95 °C, followed by 40 cycles of denaturation for 30 s 95 °C, and annealing/elongation for 34 s at 60 °C, and a final melting curve analysis”.

2- Line 196: please add the directions for the primers

Thanks, we added the directions for the primers

3- Please check the writing of bacterial species, should be italic (please see attached file)

We have done corresponding revision in whole manuscript according to your comment.

4- Please revise the references for unity according to the journal requirements (please see attached file).

According to your suggestion, we revised the references for unity according to the journal requirements

5- For other "Minor comments", please see the attached file.

As for the referee’s concern, we have done corresponding revision in whole manuscript according to your comment.

Round 2

Reviewer 1 Report

The manuscript have been improved. However, a four more remarks require improvement of the given answer.

Answer: The vehicle used in this study is silicon dioxide.

Comment: It is not a very common vehicle type. Was it obtained by microcapsulation? I suggest adding some references.

Answer: Because the tibia mid-diaphysis of bird is generally of elliptical shape, thus the mean diameter of the tibia was determined by a digital caliper.

Comment: I asked about the plane of perimeter measurements. As tibia diaphysis cross-section shows elliptical shape, It is important to give the information about the plane of measurements. Or it was meagered in two perpendicular planes and the average is reported ?

Answer: It [mucin] is intestinal barrier proteins.

Comment: Yes, mucins are intestinal barrier proteins, but not of tight junction. Please rephase.

Answer: As you known, the serum bone turnover markers just revealed that these changes in whole bone remolding, including tibia. It is hard to locate the epiphysis or diaphysis. In this regard, more precise parameters are needed to describe analyzed bone sanction in more detailed manner.

Comments: I was not asking about bone turnover markers, but about the location of trabeculae, were TRAP activity was measured and presented (Figure 6). Please also add the scale bars to Fig 6A.

Author Response

Responses to the reviewers’ comments

Dear Reviewers:

We are truly grateful to yours and reviewers’ critical comments and thoughtful suggestions. Based on these comments and suggestions we have made careful modifications on the original manuscript. All changes made to the text are in red color. Below you will find our point-by-point responses to the reviewers’ comments/ questions:
The manuscript has been improved. However, a four more remarks require improvement of the given answer.

Answer: The vehicle used in this study is silicon dioxide.

Comment: It is not a very common vehicle type. Was it obtained by microcapsulation? I suggest adding some references.

Indeed, the vehicle is not a very common type, thus we confirmed with Charoen Pokphand Group (Henan, China), and were told this just a common silicon dioxide not a Colloidal silicon dioxide, just for development new production. As for the referee’s concern, we try our best to research some similar study and add some references, unfortunately, we failed, in which, we seek for the reviewer and editor’s tolerance and understanding.

Answer: Because the tibia mid-diaphysis of bird is generally of elliptical shape, thus the mean diameter of the tibia was determined by a digital caliper.

Comment: I asked about the plane of perimeter measurements. As tibia diaphysis cross-section shows elliptical shape, it is important to give the information about the plane of measurements. Or it was meagered in two perpendicular planes and the average is reported?

We are so sorry for misunderstand. In practice, the circumference of the middle part of tibia were measured by flexible rule, therefore, we revised as following: “Tibial length and mean diameter of the tibia at mid-diaphysis (due to the mid-diaphysis of bird is generally of elliptical shape) were determined using a vernier caliper. The circumference of the middle part of tibia were measured by flexible rule”. Thanks.

Answer: It [mucin] is intestinal barrier proteins.

Comment: Yes, mucins are intestinal barrier proteins, but not of tight junction. Please rephase.

Thanks for your professional suggests. We have done corresponding revision according to this comment in whole manuscript.

Answer: As you known, the serum bone turnover markers just revealed that these changes in whole bone remolding, including tibia. It is hard to locate the epiphysis or diaphysis. In this regard, more precise parameters are needed to describe analyzed bone sanction in more detailed manner.

Comments: I was not asking about bone turnover markers, but about the location of trabeculae, where TRAP activity was measured and presented (Figure 6). Please also add the scale bars to Fig 6A

As for the referee’s concern. It located proximal tibia; Thus, we specified the location as following: “supplementation of CAR+CIN reduced the number of TRAP-positive cells in proximal tibia”.

By the way, we added the scale bars in Fig 6A
